# Administering an Appeasing Substance to Improve Performance, Neuroendocrine Stress Response, and Health of Ruminants

**DOI:** 10.3390/ani12182432

**Published:** 2022-09-15

**Authors:** Bruno I. Cappellozza, Reinaldo F. Cooke

**Affiliations:** 1Nutricorp, Araras 13601-000, SP, Brazil; 2Department of Animal Science, Texas A&M University, College Station, TX 77843, USA

**Keywords:** bovine appeasing substance, cattle, stress

## Abstract

**Simple Summary:**

Stress is present in several management activities of beef and dairy cattle, leading to health and productive losses to the herd. Therefore, strategies are warranted to reduce any losses related to these stressful situations, and bovine appeasing substance (BAS) is arising as a potential technology in livestock production settings. Several peer-reviewed publications have evaluated BAS in different production settings, such as weaning, feedlot entry, castration, transport to the slaughterhouse, and pre-weaning dairy cattle. Altogether, performance, health, and carcass traits have been positively impacted by BAS administration, demonstrating the efficacy of this technology for ruminants.

**Abstract:**

The present review demonstrates the main attributes of stress-related responses in ruminants, and the potential interaction with the immune system of the host is also presented, demonstrating that alternatives that reduce the response to stressful situations are warranted to maintain adequate health and performance of the herd. In this scenario, pheromones and their modes of action are presented, opening space to a recent technology being used for ruminants: bovine appeasing substance (BAS). This substance has been used in different species, such as swine, with positive behavioral, health, and performance results. So, its utilization in ruminants has been reported to improve performance and inflammatory-mediated responses, promoting the productivity and welfare of the livestock industry.

## 1. Introduction

Routine handling and management strategies often employed in the rearing of beef and dairy cattle may trigger stress-induced inflammatory responses, which have detrimental impacts on cattle health and performance [1], as well as the overall profitability of the livestock operation. In general, inflammatory responses triggered by stressful activities are not needed by the animal, given that a stressful response is generated by the perception of a stressor, which will push the animal away from its homeostasis [1]. Among these routine management procedures, weaning, transport, and feedlot entry are recognized as stressors that might impact productivity of the herd and can be categorized as psychological (i.e., human handling, weaning, and arrival at a novel environment), physical (i.e., castration and dehorning), and physiological (endocrine and metabolic alterations resulting from psychological and physical) stressors [2].

During their productive lives, ruminants are often exposed to various amounts and types of stressors that will likely elicit a stress response, but may not involve a pathogen itself [2]. In turn, the occurrence of stress may facilitate the potential negative effects of a specific or a group of pathogens in the body of the host animal, including those already present inside the animal that may establish a disease upon immunodeficiency. Although indispensable and needed for the resumption of homeostasis, the occurrence of a stress-induced response and its resulting inflammatory events may be unnecessary and results in negative effects on the herd’s performance and health [1], suggesting that strategies to alleviate such effects are needed. Therefore, the objective of this review is to elucidate how stress happens and triggers an inflammatory reaction in the host animal, as well as how the utilization of bovine appeasing substance (BAS) might be used as a feasible strategy to alleviate stress-related health and performance losses in livestock (for beef and dairy) production systems. It is noteworthy that an overall reduction in these losses will improve global (animal and worker) welfare, public perception on animal production, and the profitability of the entire ruminant production chain.

## 2. A Summary on Stress Physiology

From a classical standpoint, stress is defined as the reaction of an animal to factors that potentially influence its homeostasis, and animals that cannot deal with these factors are termed as stressed [3]. Stress may also be defined from a physiological standpoint as an immediate activation of the hypothalamic–pituitary–adrenal (HPA) axis following the perception of a stressor [4], characterized by the synthesis and release of corticotropin-releasing hormone (CRH) and vasopressin (VP) by the hypothalamus [2]. In cattle, CRH is more potent than VP, ultimately leading to the activation of the pituitary gland-mediated release of adrenocorticotropic hormone (ACTH) [5]. Corticotrophs located in the anterior pituitary gland are responsible for the production of ACTH, which, in turn, stimulates the synthesis and release of steroids by the adrenal gland, and promotes cholesterol uptake [2].

The cortex and medulla are the two regions of the adrenal gland, with the former synthesizing and releasing three hormone types, including the glucocorticoid called cortisol. Cortisol is well-recognized as the “stress hormone”, presenting important activities, including the (i) breakdown of body tissues, such as glycogen, muscle, and fat, to provide energy to the host during an immune challenge; (ii) production of acute-phase proteins (APPs) by the liver; (iii) regulation of a stress response; (iv) stimulating the synthesis and release of catecholamines; and (v) prevention of autoimmune disorders [2]. Hence, during chronic inflammation, cortisol remains elevated for a prolonged period of time, promoting an anti-inflammatory and immunosuppressive response [6,7], but acute increases in cortisol have been reported to trigger an acute, transient, and temporary inflammatory cascade. Besides the protective effects described above, greater cortisol concentrations have been associated with reduced growth rates and reproductive performance in ruminant females, regardless of the breed [8,9,10,11].

## 3. Stress and Immunity Interaction

The immune system is divided into a non-specific and specific immunity called innate and acquired branches, respectively. The latter is primarily responsible for adapting and building a specific immune response for each challenge the body encounters, ultimately leading to the production of antibodies and the development of immunological memory [12]. On the other hand, the innate immunity builds an acute response that is similar and independent of the type of stressor (pathogen or stressful factors) [2]. Defensive mechanisms of the innate immunity include physical and chemical barriers, as well as the complement system [12]. The main goal of innate immunity is to build a response that provides enough time for the acquired immunity to develop a strong and effective response against any specific pathogen, as well as an immunological memory in case the organism encounters the same pathogen. On a cellular level, phagocytic cells (neutrophils, monocytes, macrophages, and dendritic cells), natural killer (NK) cells, and cells that release inflammatory mediators, such as mast cells, basophils, and eosinophils, are key components of the innate immunity [12].

The innate immunity recognizes certain structures present in different microorganisms, known as pathogen-associated molecular patterns (PAMPs) [2], which start an inflammatory cascade in cattle [13,14]. Following PAMPs’ interaction with endogenous or surface Toll-like receptors (TLR) of phagocytic cells, a cytokine response is initiated in neutrophils and macrophages by the activation of the transcription factor called nuclear factor kappa beta (NFĸB) [12]. Cytokines are chemical messengers released by phagocytic cells during an immune response [15] and the major pro-inflammatory cytokines released include interleukin-6 (IL-6), IL-1, and tumor necrosis factor-α (TNF-α) [16]. Following the initial encounter with a pathogen and its deleterious effects on cellular level, the body of the host triggers a complex set of reactions that aim to restore homeostasis [2], known as the acute-phase response (APR) [17]. Tissue macrophages and blood monocytes are key cells of the APR, releasing the aforementioned cytokines (IL-1 and TNF-α) into the circulation [18]. As expected, the occurrence of an APR leads to metabolic changes in the host, such as alterations in mineral metabolism, increases in the number of white cells, as well as behavioral changes, including loss of appetite, lethargy, and decreased aggressive, sexual, and social behavior [13]. However, the two main physiological responses observed during an APR and acute inflammation are the febrile response (increased body temperature) and liver metabolism alterations [2].
(a)*Febrile response*: Key defense mechanism that controls replication and growth of pathogens, leading to its death by preventing the formation of bacterial coats. The eicosanoid prostaglandin-E2 (PGE2), produced from 20 carbon omega-6-derived fatty acids (arachidonic acid; C20:4 n-6), is the primary inducer of the febrile response. Nonetheless, it is noteworthy that the fatty acid profile of the diet might modulate such response. In other words, feeding a diet with feedstuffs containing greater concentrations of omega-6 fatty acids (i.e., linoleic acid and its derivatives) will lead to a greater pro-inflammatory state and, therefore, the occurrence of a febrile response, whereas an anti-inflammatory response is observed by feeding diets containing a greater amount of omega-3 fatty acids (i.e., linolenic acid and its derivatives) [19,20]. Although needed for controlling an infectious challenge, the increase in body temperature also results in a significant increment of nutrient requirements, so that every 1 °C increase in body temperature increases energy requirements by 10–13% [21].(b)*Hepatic metabolism*: Under homeostasis, the liver produces a wide range of APPs at a relatively steady state, but this scenario changes when the animal faces an immunologic challenge. Acute-phase proteins are a group of blood proteins that function to inhibit protease activity, repair and remodel tissues, activate macrophages, and transport proteins for products generated during an inflammatory process [22]. During an inflammatory response, the synthesis and release of APPs are stimulated by the aforementioned pro-inflammatory cytokines, and liver function is mostly directed to supply APPs [2]. Besides changes in the amount of APPs being released during an immune challenge, the profile of these released APPs is also changed [22]. In turn, the metabolizable protein (MP) requirements of the animals might also change during the period of time that APP production is hastened. Newly weaned beef calves fed 115% of daily MP requirements had a greater growth performance over a 42-day preconditioning period, demonstrating that a greater MP supply can maintain APP production with a concomitant increase in performance [23]. Some of the APPs often evaluated and of importance to ruminants (i.e., haptoglobin and ceruloplasmin) peak between 1 and 4 days post-immunological challenge, causing a significant decrease in dry matter intake (DMI) and animal performance [1]. Thus, it is not surprising that some APPs have been reported to be negatively correlated with average daily gain (ADG) and positively associated with the incidence of morbidity and antimicrobial treatments in beef cattle [24,25,26]. In ruminants, haptoglobin, ceruloplasmin, serum amyloid-A, and fibrinogen have been the most studied APPs [27], with haptoglobin being highlighted as the most reliable and consistent APP. In healthy animals, haptoglobin concentrations are nearly undetectable, but an increase up to 1000× in its concentration is often reported following the occurrence of an immune challenge, such as disease, injury, or a neuroendocrine stress response [1,24,28,29].

Unlike pathogenic challenges, in stress situations, PAMP recognition may not occur, as no pathogen is directly involved at the beginning of the inflammatory response [29,30]. Conversely, damage-associated molecular patterns (DAMPs) might be recognized following the occurrence of a stressful events. DAMPs are host biomolecules that can initiate and perpetuate a non-infectious inflammatory response in the host [31], presenting a defined intracellular function that leads to the denaturation of the protein [32]. Examples of DAMPs include DNA, RNA, mono- and polysaccharides, purine metabolites (ATP, ADP, and uric acid), and S-100 proteins [33,34,35]. In one of the first research studies aiming to evaluate the links between stress and inflammation, Cooke and colleagues (2012) developed a neuroendocrine stress model using CRH as a non-pathogenic stimulus in beef cattle [36]. Following CRH infusion, plasma cortisol concentrations peaked at 30 min, whereas TNF-α, rectal temperature, and plasma haptoglobin concentrations also increased in animals receiving 0.1 µg/kg BW of CRH, highlighting the fact stress triggers an immediate immune response, even when a pathogen is lacking [36].

## 4. Pheromones: An Overview

Every living organism uses different forms of chemical communication, which are known as semiochemical compounds. Examples of semiochemicals include the pheromones that are involved in communication among animals within the same species [36,37,38,39]. The word pheromone has a Greek origin, in a manner that ‘pheron’ means ‘to transfer’ and ‘hormon’ means ‘to excite’ [40]. To date, Karlson and Lüscher (1959) were the first researchers that defined pheromones as “substances secreted by a specific individual to the outside and received by a second individual from the same species, triggering a specific reaction, such as a behavior and/or a development process” [41]. Nowadays, pheromone may also be defined as “a substance that is used as an intraspecific communication tool that does not elicit apparent behavioral or endocrine alterations” [39,40]. Moreover, a pheromone must be released to the outside of the body and perceived by cohorts from the same species, while the reactions in the individuals must be inborn [42].

Pheromones are chemically classified as organic, hydrophobic, and volatile molecules [43], generally composed of one or more active compounds. From a structural standpoint, pheromones are relatively simple (i.e., hydrocarbons) or more complex [44,45], being naturally or endogenously produced in vaginal secretions, feces, saliva, and urine [46]. The utilization of pheromones is known in several species, such as rodents and arthropods, whereas, in ruminants, most of the focus has been on the reproductive function of males and females [40]. Humans have been using pheromones from rodents and arthropods as a manner to control their replication in urban settings [40]. Hence, the identification of specific pheromones of interest will lead to the development of synthetic analogues that might benefit the population and/or a specific segment. Overall, pheromones can be classified as attractant [39], repellent (i.e., citronella) [39], recognition [40,47], signal [40], aggregation [48], and sexual [40].

### Mode of Action

Overall, pheromones are detected and conducted to target mammalian organs, in a mechanism classified as “Flehmen Reflex” [49]. In ruminants, target organs involved in pheromone perception include the main olfactory epithelium (MOE) and vomeronasal organ (VNO) [40]. The first is responsible for the recognition of traditional odor molecules and chemical environmental signals without specificity or meaning, whereas VNO is related to pheromone recognition, carrying specific intraspecies chemosensory signals [50] through the receptors [51], leading to the occurrence of a neuroendocrine cascade independent of an animal’s cognitive recognition [46]. Nonetheless, others have demonstrated that both systems (MOE and VNO) along with other olfactory organs are involved in pheromone detection [52,53]. The VNO is located between the mouth and nose of mammals and, in ruminants, Fukuda et al. (2009) reported the presence of an odorant-binding protein (OBP), termed as bovine colostral OBP (bcOBP), that acts as a pheromone transporter [54,55,56] based on the evidence that maternal pheromones could bind to the porcine colostrum–milk OBP [57]. Therefore, OBPs in maternal fluids (i.e., colostrum and milk) were assumed to be involved in the recognition of the mother by her neonates [57].

The VNO neurons can encode stimulus strength, activating an entire neural subpopulation and conducting an electrochemical signal to the mammalian brain [40], stimulating the hypothalamus to exhibit an appropriate neuroendocrine response unique to the specific subpopulation of neurons stimulated in the VNO. In fact, the effects of pheromones on reproduction occur primarily due to the release of gonadotropin-releasing hormone (GnRH) by the hypothalamus. In sheep, the presence of a male can stimulate an LH surge in an anestrous female, whereas the rhythm of LH pulse release remains elevated for several hours and the pre-ovulatory surge is observed approximately 36 h following male exposure [58], stimulating the ovarian secretion of estradiol that, via positive feedback at the hypothalamic level, induces a pre-ovulatory LH surge. Therefore, pheromone perception plays a key role in the reproductive function of the herd, allowing mate identification and an adequate response [59]. Research in male wild pigs demonstrated that damage to VNO resulted in loss of the ability to breed a female wild pig. In females of the same species, there was a lack of interest in a sexual partner and pregnancy became rare [60]. In small ruminants, the posterior part of VNO contains a sensorial epithelium that allows the perception of a non-volatile molecule through the oral cavity of the organ, causing the “Flehmen Reflex” [61]. In ruminants, the dam–offspring bond formation is mediated by olfactory signals and VNO [62], but few studies have directly focused on cattle and how these organs interact following a pheromone perception.

## 5. Pheromones in Livestock Production

In the last 25 years, increased interest in pheromone action and its effects has been observed in several species, such as cattle [63,64], pig [58,65,66,67], horse [68], goat [69], and sheep [70]. In these species, several benefits associated with natural pheromones have been reported for social behavior and reproductive function. Therefore, the development and subsequent utilization of synthetic analogues of endogenously released compounds are expected to benefit livestock production.

Swine has been one of the main species focused on for the effects of pheromone utilization on behavior and performance of the flock. Indeed, maternal pheromones regulate nursing pig behavior in a manner that the lack of a specific odor from the skin of the sow leads to recognition and discrimination among maternal odors found in feces and other biological fluids, for example [71,72]. A decade later, Pageat (2001) was able to isolate secretions from the skin of mammals, leading to the development of a synthetic analogue similar to the endogenous secretion observed in the sow skin [67]. In one of the first trials evaluating this technology, Pageat and Teissier (1998) demonstrated that pig aggressive biting behavior was reduced following weaning and commingling [73]. Moreover, pigs treated with the synthetic substance at weaning presented altered behavior, including more time feeding and standing/walking, reduced amount of time drinking water, lying down, and engaging in agonistic behaviors [74]. Over a 4-week period, treated pigs were heavier (+1 kg), had a greater ADG (23%), and feed efficiency (FE; 22%) vs. placebo-treated pigs [74]. These data demonstrated that the synthetic pheromone analogue caused behavioral changes that positively affected post-weaning performance of the sounder.

In support of the aforementioned results in swine, a fair and valid rationale would be to evaluate the effects of an appeasing substance in ruminants, as these experience different stressful situations, such as weaning, transport, feedlot entry, and castration [2]. Therefore, the application of a bovine appeasing substance (BAS; IRSEA Group, Quartier Salignan, France) has been evaluated by our and other research groups. For the current and upcoming sections, it is important to briefly state that BAS is based on a proprietary mixture of fatty acids including saturated (palmitic and oleic) and polyunsaturated (linoleic), which are added at 1% of the excipient, yielding a long-lasting effect of 15 days in the treated animal [65,66,67]. Furthermore, BAS reproduces the components of the natural pheromone produced by beef and dairy females [65,66,67].

### 5.1. BAS vs. Weaning

Among the several stressful events faced by the ruminants, weaning is highlighted [1] and management alternatives to alleviate the stress-related losses following weaning have been addressed by other researchers [75,76,77]. Nonetheless, technologies that could be employed at the moment that weaning occurs are still warranted, as most of the beneficial ones involve management plans and/or activities that production systems might not be able to apply in their daily operations. Therefore, our research group has evaluated the utilization of BAS in different weaning settings and how this technology might be able to impact the physiological responses, health, and productive function of the beef cattle herd.

Cooke and colleagues (2020) administered a single dose of BAS to *B. indicus* × *B. taurus* beef calves at weaning (5 mL/head) and evaluated their performance for a 45-day period [78]. These authors observed that BAS-treated calves had a greater ADG (+70 g/day) and final BW (+2.8 kg) when compared to non-treated cohorts (*p* = 0.03; Table 1). Following a similar experimental design, *Bos indicus* beef calves receiving BAS at the moment of weaning gained more BW and were heavier at the end of the 45-day experimental period vs. control cohorts (Table 1) [79]. In a subsequent study, Schubach et al. (2020) also demonstrated that behavior of *B. taurus* calves was greatly impacted following BAS administration at weaning, such as temperament, feeding, and allogrooming behaviors [80].

One of the mechanisms by which BAS improves performance might be related to a less heightened neuroendocrine stress response. According to the aforementioned inflammatory cascade, BAS-administered beef steers had a reduced mean hair cortisol 14 days post-weaning compared with the non-treated group (Figure 1) [80] as well as reduced mean haptoglobin concentration during the post-weaning period (Table 2) [78,80]. Stress also affects the efficacy by which a specific vaccine is able to induce an inflammatory response and the body to mount an effective and robust immunological memory [1,81]. Then, it would be feasible to speculate that an alleviated neuroendocrine stress response post-BAS administration could improve the efficacy of a specific vaccine. Indeed, calves that were vaccinated and received BAS at weaning had a greater concentration of antibodies against bovine respiratory disease (BRD) [80].

### 5.2. BAS vs. Feedlot Entry

In particular, feedlot arrival and entry are simply the tip of the iceberg when we take into consideration all the stressors that an animal might be exposed to. For example, the animal might be managed in the working chute while still in the farm of origin, handled by humans, vaccinated, ear-tagged, castrated (if applied), loaded in a commercial truck, transported, restricted from feed and water, experienced environmental changes, unloaded in a novel environment, novel management, processing, commingled with a different group of animals from a different source/origin, and received novel sources of feed and water, among others. Under a production setting, these stressful situations might occur over a 24–48 h period, stimulating a neuroendocrine stress response at feedlot entry. Moreover, it is not surprising that BRD incidence is elevated in the beginning of the feedlot period (14–21 days), as the immune system is suppressed and BRD pathogens are able to establish the disease [76,81]. Therefore, BAS administration may be an alternative to alleviate these neuroendocrine responses.

Following this rationale, administering BAS at feedlot entry improved the performance of *B. indicus* bulls in the first 15 days of the feedlot, whereas no benefits were observed in the whole 45 days of the experiment (Table 3) [78]. The lack of effects following BAS administration is unknown, but could be related to a compensatory gain in the non-treated group and/or the need for other BAS applications during the feedlot period. Nonetheless, in a subsequent study, Colombo and colleagues (2020) evaluated how BAS administration to cattle at feedlot arrival would affect their performance and health [82]. Cattle used by Colombo et al. (2020) were purchased in an auction facility and originated from 16 different ranches, to mimic the stress load that commercial cattle are often exposed to [82]. Administration of BAS reduced plasma cortisol and increased plasma glucose concentrations 7 days post-feedlot entry, indicating a reduced stress response and a greater nutritional status of the herd following feedlot entry, respectively. Over the initial 45 days of feedlot, ADG (0.857 vs. 1.013 kg/day) and FE (142 vs. 171 g/kg) were also improved due to BAS administration (Table 3), supporting the rationale that a stress reduction in the beginning of a specific period (stocking and/or feedlot entry) positively impacts the performance and health of the beef cattle herd. This rationale has already been established and reported for dairy cattle in the transition/post-calving period [83,84], but few data are available in beef cattle.

An additional question that remained was whether the timing of BAS administration, pre- or post-transport, and if administering two doses, pre- and post-transport, would lead to different results in the entire feedlot period (108-day feeding period). Based on this rationale, Fonseca et al. (2021) designed a trial in which BAS was administered (1) pre-transport to a commercial feedlot (at loading); (2) post-transport, during the initial processing management at the commercial feedlot (at feedlot entry); (3) pre- and post-transport; and (4) no BAS administration [85]. Administration of BAS at loading benefited animal performance during adaptation (19 days), tended to increase ADG during the finishing period, and improved overall ADG. As a result, bulls were heavier at the end of the experimental period (108 days) and also had a heavier carcass (Table 4) [85].

These results highlight, once more, how BAS might be able to improve the overall performance and carcass characteristics of beef cattle during finishing, a period where several stressors are observed and have been recognized to negatively impact the health and productivity of the herd [1]. Based on the results above [85], it can be argued that the main mechanism by which BAS improves herd performance is on nutrient utilization, denoted by the FE results, a trait greatly impacted in stressed animals [1,86]. In fact, a stressor, when perceived by the animal, might lead to the inflammatory cascade that acts as nutrient sink, removing the nutrients from an anabolic to a catabolic state [16]. Moreover, blood mRNA expression of pro-inflammatory-related genes was reduced following weaning and feedlot entry in animals that received BAS [87], demonstrating that the modulation of inflammation might be one of the mechanisms underlying the benefits observed by [85].

On the other hand, BAS administration at feedlot entry yielded immediate benefits during the adaptation phase (19 days), but failed to promote long-term effects on performance over 108 days. One might argue that the lack of long-term effects is due to the fact that most of the stressful situations had occurred at feedlot entry, triggering an APR, and likely overriding the benefits of BAS in finishing animals [88,89]. Lastly, the same rationale might be used to explain the lack of positive effects when two applications of BAS had been performed, at loading and feedlot entry [85].

### 5.3. BAS vs. Castration

Castration is a widely adopted management practice performed in feedlot cattle that eliminates aggressive behavior in male animals due to the removal of endogenous testosterone production, while also improving backfat thickness and meat characteristics [90,91]. On the other hand, castrated animals often present a reduced overall ADG, final BW, FE, HCW, and ribeye area (REA) following the feedlot phase and subsequent slaughter [92]. In the U.S., it is estimated that 93% of the animals arriving at the feedlot are already castrated, being 50 and 43% surgically and band-castrated [93], whereas an estimated age at which this procedure occurs is unknown. In developing countries such as Brazil, castration is mostly performed in operations that attend a prime niche beef market, such as restaurants and butcheries that sell a specific beef brand. Nonetheless, regardless of the region we focus our attention, due to public opinion, animal production, practices, care, and consequently welfare have been under scrutiny. Recent surveys indicated that public opinion is more positive when castration is performed with any kind of anesthesia and/or analgesia [94].

The castration process itself also causes physical, physiological, and psychological stress in animals, resulting in inflammatory reactions and subsequent performance loss that might last until slaughter [95]. One alternative to alleviate these immune responses and consequently maintain an adequate performance post-castration is the administration of meloxicam, a non-steroidal anti-inflammatory drug (NSAID) [96]. Nonetheless, vaccination and/or intravenous or intramuscular pharmacological administration might also cause an immune response due to the resulting local tissue injury [17,97] and the industry itself has been avoiding the use of medicines in animals. Hence, technologies that do not cause a local tissue injury, immune response, and improve the performance of newly castrated feedlot animals are warranted.

Based on this rationale, our research group designed an experiment [98] to evaluate the effects of administering BAS at castration on performance of Nellore × Angus calves (*n* = 390; initial BW = 274 ± 21.0 kg). On day 0 of the study, individual calf BW was recorded and animals were assigned to receive BAS (*n* = 195; 5 mL/animal) or placebo (CON; *n* = 195; 5 mL/animal). Immediately after treatment administration, all calves were castrated using a burdizzo by trained feedlot personnel. During the feedlot entry period (days 0 to 30), all animals were offered a 60:40 roughage:concentrate diet based on grass hay, whereas, during the finishing period (days 31 to 258), a high-concentrate diet was offered. On day 30 of the study, BAS-administered animals were heavier and had a greater ADG than CON cohorts (Table 5). Similarly, BW change and ADG were also greater for BAS vs. CON on day 258 (Table 5). As a descriptive analysis, total DMI from days 0 to 30 of the study was 6.70 kg/d for CON and 6.75 kg/d for BAS, resulting in a greater numerical FE for BAS (146 vs. 172 g/kg, respectively). Overall, from days 0 to 258, DMI and FE were 7.58 vs. 7.59 kg/d and 141 vs. 146 g/kg for CON and BAS, respectively, whereas hot carcass weight tended to be greater for BAS (Table 5). These data indicate that the heavier BW was translated into a heavier carcass, which is the main parameter that determines the economic profit of a feedlot operation. The increased ADG during the first 30 days following castration in BAS-administered beef animals might be related to reduced stress-induced physiological and inflammatory reactions known to impair cattle BW gain, such as the acute-phase protein response [78]. In an ideal production setting, it would be interesting to investigate whether BAS administration at castration maintains similar performance when compared to non-castrated beef animals. Although positive results have been reported in beef cattle following different stressful procedures, more research is warranted to understand the underlying biological mechanisms, if more than reported by [87], by which BAS promotes an improvement in the performance and health of the beef cattle herd, more specifically in different stressor types. The potential benefits of administering BAS under a pathogen challenge are also worth mentioning, as this might be tightly connected to stressful situations, such as transport/feedlot entry and the occurrence of the bovine respiratory disease complex in the first 30 days on feed.

In summary, BAS administration at castration improved ADG and BW change 30 days following castration, whereas these positive results persisted throughout the entire feedlot period. Additionally, hot carcass weight was also greater for BAS-administered vs. CON cohorts, demonstrating that BAS is a feasible technology to improve the performance and carcass traits of feedlot cattle.

### 5.4. BAS vs. Dark, Firm, and Dry Cuts

Prior to slaughter, animals face physical, psychological, and physiological stressors, increasing the chance of the occurrence of dark, firm, and dry (DFD) carcasses. These stressors and the resulting neuroendocrine stress response ultimately alter meat quality and customer acceptance of this edible product, particularly due to an increase in meat pH and changes in meat tenderness and color [99]. The pH of the carcass is greatly affected by the total and rate of glycogen breakdown, which is impacted by acute and chronic stress [100]. More specifically, pre-slaughter stressors might stimulate ATP reduction, muscle glycogen depletion, and alterations in important physical and chemical attributes of the meat [101,102,103]. As a general definition, meat pH values greater than 5.80 from 12 to 48 h postmortem will yield DFD cuts, which, in turn, become more susceptible to microbial contamination and reduce shelf-life [99,103]. These changes are often associated with a reduction in product acceptance by customers, as DFD cuts are dark red to brown-black and have a dry, firm, and sticky consistency [104]. Meat traits that have greater influence on consumer satisfaction are tenderness, juiciness, and flavor of the cooked meat, demonstrating the reason why DFD cuts are less accepted [101].

It is important to mention that other factors also predispose carcasses to DFD, such as sex, breed, nutrition, animal category, temperament, and age. In different countries, DFD occurrence has been reported to be in the range of 2.0–13.5% [105,106]. Considering that *B. indicus* breeds are more temperamental than *B. taurus* [107], it is reasonable to speculate that *B. indicus* animals would have a heightened stress and APR response, resulting in a greater occurrence of DFD carcasses. Hence, technologies that alleviate neuroendocrine stress responses and improve carcass quality are warranted. In order to assess the efficacy of BAS in preventing a postmortem increase in meat pH, Cappellozza et al. [79] applied BAS (5 mL per head) at the moment animals were being loaded into commercial trucks and transported to the slaughter plant. These authors reported that BAS administration at loading was effective for maintaining mean carcass pH below the 5.80 threshold and reduced the proportion of carcasses that were classified with a mean pH greater than 5.80 (Table 6) and 6.00 (19.4 vs. 11.2% of the carcasses).

### 5.5. BAS vs. Dairy Cattle

The dairy production system is characterized by intensive management situations, including daily milking, weaning, changes in nutritional management, prepartum DMI reduction, late lactation dry-off, vaccination program, reproductive management, novel environment depending on the season and days on milk, as well as commingling with different animals on different days and stages of lactation that alter social hierarchy of the herd [108,109,110]. These situations might predispose the animals to stress and consequently impact overall health, performance, and herd longevity.

In one of the first studies in dairy cattle, weekly BAS administration to lactating dairy cows at the moment of turn out to pasture resulted in greater milk production (+1.65 kg/day) and a reduction in somatic cell counts (13.0%), supporting the statement that BAS reduced the stress-related response of the animals due to an environmental change [111]. In agreement with these data, an immediate increase in SCC was reported when lactating dairy cows were moved to a pasture setting [112]. Recently, Angeli et al. [113] addressed the effects of 14-day BAS administration on performance and health traits of pre-weaning Gir × Holstein dairy female calves. Bovine appeasing substance administration did not impact disease occurrence, but it tended to decrease the days of pharmacological intervention and reduced the cost of the pharmacological interventions (Table 7). Additionally, BAS-administered calves were heavier at weaning (+3.8 kg) mainly due to greater ADG from days 42–56 and 56–weaning (Table 7). Another interesting finding reported by Angeli and colleagues [113] was the fact that BAS-administered animals diagnosed with a disease had greater ADG than control animals diagnosed with a disease and similar ADG vs. healthy control animals. This was the first study demonstrating that BAS administration might be able to recover the animals at a faster rate following a pathogen challenge and, consequently, improve pre-weaning performance. Additionally, the benefits on pathogen-challenged animals are novel and different from other technologies, such as NSAID, that have improved productivity in animals challenged with the neuroendocrine stress model [86,92], but the same effects were not observed upon an LPS or vaccine challenge [14].

## 6. Conclusions

Stress inevitably occurs in daily management situations faced by beef and dairy animals throughout their productive lives, leading to performance and health losses that can significantly impact the overall productivity of the herd. Therefore, alternatives that reduce these losses and improve animal welfare are warranted and should be further evaluated. Such an alternative might be the pheromones, chemical signals produced by the animals that might positively benefit another animal, either by promoting a specific trait (reproduction, production, and health) or signaling a dangerous situation. More specifically, the identification, isolation, and synthesis of synthetic analogues might be able to translate these natural endogenously produced substances into a commercial application, including bovine appeasing substance (BAS). Recently, BAS has been evaluated in different commercial settings of beef and dairy cattle, with significant improvements for the performance and health of the herd, following an encounter with a stressful situation, such as pre-weaning, weaning, feedlot entry, castration, and slaughter. Additional research efforts should be considered and are encouraged to further elucidate potential underlying modes of action by which BAS benefits the performance and health of the beef and dairy cattle herd, as well as novel situations in which this technology might improve the health, performance, and welfare of the herd.

## Figures and Tables

**Figure 1 animals-12-02432-f001:**
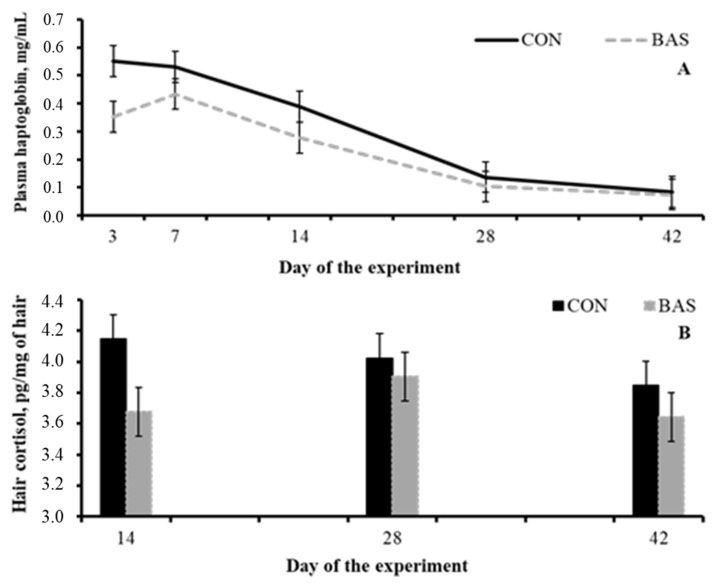
Plasma haptoglobin concentrations (**A**) and hair cortisol from the tail switch (**B**) from beef calves receiving (BAS) or not (CON) a bovine appeasing substance at weaning (day 0). Adapted from Schubach et al. [80].

**Table 1 animals-12-02432-t001:** Mean post-weaning performance of newly weaned calves receiving (BAS) or not (CON) a bovine appeasing substance at weaning ^1,2^.

Item	BW, kg	ADG, kg
Cooke et al. (2020)		
CON	248.6 ^b^	0.29 ^b^
BAS	251.4 ^a^	0.36 ^a^
Cappellozza et al. (2020)		
CON	240.3 ^b^	1.08 ^b^
BAS	256.5 ^a^	1.45 ^a^
Schubach et al. (2020)		
CON	228.4	1.04
BAS	230.6	1.08

^1^ Results were adapted from published articles [78,79,80]. ^2^ Different letters denote significance at *p* < 0.05 level.

**Table 2 animals-12-02432-t002:** Physiological responses of newly weaned beef steers receiving (BAS) or not (CON) a bovine appeasing substance at weaning ^1,2^.

Item	CON	BAS
Haptoglobin, mg/dL		
Weaning	0.395	0.395
Day 15	0.530 ^a^	0.279 ^b^
Day 45	0.246	0.236
Hair cortisol, pg/mg of hair		
Weaning	2.48	2.48
Day 15	2.47	2.51
Day 45	2.55	2.59

^1^ Results were adapted from [78]. ^2^ Different letters denote significance at *p* < 0.05 level.

**Table 3 animals-12-02432-t003:** Mean post-feedlot entry performance in beef cattle receiving (BAS) or not (CON) a bovine appeasing substance ^1,2^.

Item	BW, kg	ADG, kg	DMI, kg/d ^3^	FE, g/kg ^3^
Cooke et al. (2020)				
CON	403.9	1.58	--	--
BAS	399.8	1.50	--	--
Colombo et al. (2020)				
CON	291.1	0.86 ^b^	4.95	142 ^b^
BAS	294.7	1.01 ^a^	4.98	171 ^a^

^1^ Results were adapted from published articles [78,82]. ^2^ Different letters denote significance at *p* < 0.05 level. ^3^ DMI = dry matter intake; FE = feed efficiency. Both variables were not reported by [78].

**Table 4 animals-12-02432-t004:** Effects of administering (BAS) or not (CON) bovine appeasing substance to beef bulls at loading and/or at feedlot entry ^1,2^.

Item	Loading		Feedlot Entry			*p* = ^3^	
	CON	BAS	CON	BAS	L	F	L × F
Body weight, kg							
Day-2	341.2	341.1	341.1	341.2	0.97	0.98	0.99
Day 0	302.8	300.0	302.3	300.5	0.44	0.62	0.97
Day 19	336.1	342.0	337.3	340.8	0.15	0.40	0.61
Day 60	395.4	405.4	399.3	402.1	0.06	0.59	0.75
Day 108	457.1	471.3	461.8	466.6	0.03	0.46	0.98
Average daily gain, kg							
Days 0–19	1.798	2.210	1.880	2.119	<0.0001	0.02	0.44
Days 19–60	1.447	1.548	1.516	1.479	0.16	0.59	0.67
Days 60–108	1.274	1.372	1.301	1.345	0.10	0.45	0.53
Overall	1.430	1.586	1.483	1.553	<0.001	0.24	0.80
Dry matter intake, kg/d							
Days 0–19	6.52	6.90	6.53	6.88	0.03	0.04	0.88
Days 19–60	9.50	9.76	9.40	9.86	0.27	0.07	0.30
Days 60–108	9.48	9.93	9.53	9.88	0.07	0.14	0.30
Overall	8.97	9.33	8.95	9.35	0.08	0.06	0.29
Feed efficiency, g/kg							
Days 0–19	274	321	287	307	<0.001	0.05	0.14
Days 19–60	154	159	163	150	0.58	0.18	1.00
Days 60–108	135	149	137	136	0.39	0.79	0.97
Overall	160	170	167	164	0.05	0.57	0.67
Carcass traits							
Hot carcass weight, kg	260.1	268.9	263.1	266.1	0.03	0.46	0.93
Dressing percent	56.88	57.03	56.93	56.99	0.03	0.40	0.93

^1^ Adapted from Fonseca et al. [85]. ^2^ Day 2 = loading; Day 0 = feedlot entry; Day 19 = end of the adaptation period; Day 60 = end of the growing period; Day 108 = end of the finishing period. ^3^ L = Effects of treatment at loading (d-2); F = Effects of treatment at feedlot entry (d 0); L × F = loading × feedlot entry interaction.

**Table 5 animals-12-02432-t005:** Performance of crossbred beef steers receiving (BAS; *n* = 195) or not (CON; *n* = 195) a bovine appeasing substance at castration (day 0) ^1^.

Item	CON	BAS, kg
Body weight, kg		
Day 30	284.7 ^b^	289.6 ^a^
Day 258	531.0 ^b^	540.8 ^a^
Body weight change, kg		
Days 0–30	29.6 ^b^	34.7 ^a^
Days 30–258	245.7	251.0
Days 0–258	275.4 ^b^	286.0 ^a^
Average daily gain, kg		
Days 0–30	0.991 ^b^	1.157 ^a^
Days 30–258	1.078	1.101
Days 0–258	1.068 ^b^	1.109 ^a^
Carcass traits		
Hot carcass weight, kg	296.4	300.1
Dressing percent	55.8	55.7

^1^ Different letters denote significance at *p* < 0.05 level. Adapted from [98].

**Table 6 animals-12-02432-t006:** Effects of bovine appeasing substance (BAS) administration meat pH and proportion of carcasses with a pH greater than 5.80 in finishing beef bulls ^1,2^.

Item	CON	BAS
Meat pH	5.82 ^a^	5.75 ^b^
% carcass pH > 5.80	42.2 ^a^	26.2 ^b^

^1^ Results were adapted from [79]. ^2^ Different letters denote significance at *p* < 0.05 level.

**Table 7 animals-12-02432-t007:** Days of pharmacological intervention, cost per pharmacological intervention, and performance of female Gir × Holstein dairy calves receiving (BAS) or not (CON) a bovine appeasing substance every 14 days during pre-weaning ^1,2^.

Item	CON	BAS
Health		
Days of pharmacological intervention, days		
Diarrhea	2.2	1.9
Pneumonia	3.3	3.2
Diarrhea + pneumonia	4.7	2.9
Cost of pharmacological intervention, USD		
Diarrhea	1.91 ^a^	1.11 ^b^
Pneumonia	4.24	3.93
Diarrhea + pneumonia	6.20 ^a^	4.28 ^b^
Performance		
Weaning BW, kg	90.8	94.6
ADG, kg	0.73	0.78

^1^ Results were adapted from [113]. ^2^ Different letters denote significance at *p* < 0.05 level.

## Data Availability

The data presented in this study are available in this article.

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
