# Peer review of "Administering an Appeasing Substance to Improve Performance, Neuroendocrine Stress Response, and Health of Ruminants"

_animals, 2022, doi:10.3390/ani12182432_

Round 1
Reviewer 1 Report
Dear authors,
I was asked to review your manuscript" Administration of an appeasing substance to improve performance, neuroendocrine stress response, and health of cattle". Although beeing classified a review, I still feel, that all information on BAS comming from your own group. Maybe you are the only ones doing research on this topic. I found the article interesting and a good summary on this topic. For me it is more a book chapter and that is why I do not recomment it for publication in animals. But decisions lies with the editor.
Author Response
We would like to acknowledge the reviewer comments the quality of the manuscript and, indeed, our research group has been leading all the efforts with BAS in ruminants and, therefore, most of the references are originated from our group. Nonetheless, as mentioned to the Editor above, we also included all the relevant reference from other researchers in the ruminant (Osella et al., 2018; Hervet et al., 2021) and swine (Pageat & Teissier, 1998; McGlove and Anderson, 2002) segment.
Lastly, we structured the review manuscript as it usually is structured in the present and different journals, introducing a specific topic and then, highlighting the major topic of the manuscript, which was pheromones, potential modes of action, and BAS in different ruminant production settings.
Reviewer 2 Report
Dear Authors,
The topic of the present article (a review of bovine appeasing substance (BAS) on the performance and health of beef and dairy cattle in different stressful situations, such as weaning, feedlot entry, castration, and slaughter) is of great novelty and interest.
However, besides a few minor aspects that could be addressed to improve the overall exposition and that will be reported below, I have only one major concern. The article is presented as a Review and organized as such. However, in the section “5.3. BAS vs. Castration” at line 436 there appears to be a shift in the manuscript type. Here, the manuscript focuses on a personal experiment of the Authors, starting with the explanation of the materials and methods of the research, with no reference to this being a previously published work. I don’t understand if this is a novel unpublished work or a published pre-existing work that was not cited. However, I suggest the Authors be consistent with their manuscript type, and reference all the cited work, or alternatively, if they want to make this an original research manuscript, change the manuscript type and outline.
Besides this, my minor considerations are:
Line 30: The sentence is not quite clear, it seems something is missing, could it have been “Regular handling and management strategies” maybe?
Line 33: “This stress-induced inflammatory response is not needed by the animal” physiologically this could be a necessary and an important response in the animal that in certain contexts is needed, I suggest rephrasing the sentence for better clarity.
Lines 42 – 44 “In turn, occurrence of stress may facilitate the action and, therefore, the negative effects of an individual or a group of pathogens in the body of the host animal, including those already inoculated in the body of the host that establish diseases upon immunodeficiency.” This sentence could be rephrased to make it clearer, especially that last “including those already inoculated” because the verb inoculated suggests that they have been actively inserted in the animal by someone.
Line 87 “On the other hand, the innate immunity builds a similar response, regardless the type of stressor” I suggest rephrasing this sentence. You started explaining acquired immunity and then you moved on to the innate immunity saying it builds a “similar response”. For how it is presented here it looks like innate immunity builds a similar response to acquired immunity.
Line 106 – I suggest removing “animal” because it’s unnecessary and redundant
Line 116 – “well-known as a defense mechanism that controls replication and growth” I suggest rephrasing
Line 121 I suggest removing “with feedstuffs” and “containing feedstuffs” because it’s unnecessary and redundant.
Line 210 “compounds that work as defense mechanism against predators, selection of sexual partners” it’s not very clear, maybe this should be better at the end of the paragraph after the explanation, or maybe it could be phrased differently.
From lines 285-207 to lines 288-and on there appears to be a slight inconsistency, you were describing the effect of pheromones on weaning pigs, and then you are examining utilization of BAS on stressed cattle but it seems to me that a brief explanatory connection between the two could be necessary.
Author Response
Comment: The topic of the present article (a review of bovine appeasing substance (BAS) on the performance and health of beef and dairy cattle in different stressful situations, such as weaning, feedlot entry, castration, and slaughter) is of great novelty and interest.
Response: We would like to acknowledge the reviewer comments regarding the overall content of the manuscript.
Comment: However, besides a few minor aspects that could be addressed to improve the overall exposition and that will be reported below, I have only one major concern. The article is presented as a Review and organized as such. However, in the section “5.3. BAS vs. Castration” at line 436 there appears to be a shift in the manuscript type. Here, the manuscript focuses on a personal experiment of the Authors, starting with the explanation of the materials and methods of the research, with no reference to this being a previously published work. I don’t understand if this is a novel unpublished work or a published pre-existing work that was not cited. However, I suggest the Authors be consistent with their manuscript type, and reference all the cited work, or alternatively, if they want to make this an original research manuscript, change the manuscript type and outline.
Response: The reviewer brings a good point here. The work presented in the section 5.3 is novel and have not been presented in other articles. Considering that this is a review manuscript and this work had been done in a specific production setting of interest in different regions of the globe (castrated animals) that we realize that leads to a stressful response in the animals, the work is a great fit in the current manuscript. We have highlighted that the work has not been published before and the description has been adjusted and shortened.
Line 30: The sentence is not quite clear, it seems something is missing, could it have been “Regular handling and management strategies” maybe?
Response: Changed to “Routine handling”
Line 33: “This stress-induced inflammatory response is not needed by the animal” physiologically this could be a necessary and an important response in the animal that in certain contexts is needed, I suggest rephrasing the sentence for better clarity.
Response: The context of the statement was for inflammatory responses caused by stressful situations, like weaning and feedlot entry, for example. For better clarity, the statement was rewritten as “In general, inflammatory responses triggered by stressful activities are not needed by the animal”
Lines 42 – 44 “In turn, occurrence of stress may facilitate the action and, therefore, the negative effects of an individual or a group of pathogens in the body of the host animal, including those already inoculated in the body of the host that establish diseases upon immunodeficiency.” This sentence could be rephrased to make it clearer, especially that last “including those already inoculated” because the verb inoculated suggests that they have been actively inserted in the animal by someone.
Response: Rewritten as “In turn, occurrence of stress may facilitate the potential negative effects of a specific or a group of pathogens in the body of the host animal, including those already present inside the animal that may establish a disease upon immunodeficiency.”
Line 87 “On the other hand, the innate immunity builds a similar response, regardless the type of stressor” I suggest rephrasing this sentence. You started explaining acquired immunity and then you moved on to the innate immunity saying it builds a “similar response”. For how it is presented here it looks like innate immunity builds a similar response to acquired immunity.
Response: Rewritten as “On the other hand, the innate immunity builds an acute response that is similar and independent of the type of stressor (pathogen or stressful factors)”
Line 106 – I suggest removing “animal” because it’s unnecessary and redundant
Response: Removed as requested.
Line 116 – “well-known as a defense mechanism that controls replication and growth” I suggest rephrasing
Response: Rewritten as “key defense mechanism that controls replication and growth of the pathogen, leading to its death by preventing the formation of bacterial coats.”
Line 121 I suggest removing “with feedstuffs” and “containing feedstuffs” because it’s unnecessary and redundant.
Response: Removed as requested.
Line 210 “compounds that work as defense mechanism against predators, selection of sexual partners” it’s not very clear, maybe this should be better at the end of the paragraph after the explanation, or maybe it could be phrased differently.
Response: This section has been modified as requested by reviewers 2 and 3.
From lines 285-207 to lines 288-and on there appears to be a slight inconsistency, you were describing the effect of pheromones on weaning pigs, and then you are examining utilization of BAS on stressed cattle but it seems to me that a brief explanatory connection between the two could be necessary.
Response: Rewritten as “In support of the aforementioned results in swine, a fair and valid rationale would be to evaluate the effects of an appeasing substance in ruminants, as these experience different stressful situations, such as weaning, transport, feedlot entry, and castration [2]. Therefore, the application of a bovine appeasing substance (BAS; IRSEA Group, Quartier Salignan, France) has been evaluated by our and other research groups.”
Reviewer 3 Report
Introduction- sufficient, Objectives are clear
Better to explain the procedure of literature review. Screening of manuscripts, Time period etc.
Overview of Pheromones- too much information. Please focus on your objectives.
You should clearly introduce what is bovine appeasing substance. You should explain its' consequences according to your objectives of the manuscript.
Please check all the abbreviations used- you need to introduce them at the very first stage.
Table 1- Please rearrange the table ( CON and BAS should come as columns.) this will enhance the readability- better to use the format of Table 2
Line 320- Please specify the animal here - Beef cattle?
Line 322-324- not clear. Please rewrite elaborating the content.
Table 2- Title should be revised- Please specify the beef animal - Beef cattle is better than beef animals
Table 3- rearrange the table as in Table 2 to enhance the readability. The title should be revised, please specify the animal as beef cattle. Define DMI and FE here.
Line 374- You should mention the species here - Beef cattle
Table 4- Please include the species in the title
Line 393- better to mention as beef cattle
Line 407- Define APR
BAS vs Castration- too much information on the experimental methodology ( unpublished data of your own research team). Better to include other references also here.
Table 5- include the species in the title
Table 6- Include the species in the title
In overall, too much of information in the introduction. Better to focus on your objectives and rearrange the introduction. Better to include more related literature to support your conclusion.
Better to organize the information in a more logical sequence to support your conclusion.
Author Response
Introduction- sufficient, Objectives are clear
Response: Thanks for your comments!
Comment: Better to explain the procedure of literature review. Screening of manuscripts, Time period etc.
Response: We respect the reviewer comments, but this is not a meta-analysis or meta-regression, and evaluating the recent review manuscripts published in Animals, our structure seems to be appropriate.
Overview of Pheromones - too much information. Please focus on your objectives.
Response: It has been rewritten and shortened.
Comment: You should clearly introduce what is bovine appeasing substance. You should explain its' consequences according to your objectives of the manuscript.
Response: It has been done as suggested by the reviewer, by presenting the basics of the work done in swine (line 272-286), followed by an introduction of the BAS and potential benefits in cattle (line 287-296).
Comment: Please check all the abbreviations used- you need to introduce them at the very first stage.
Response: Checked and corrected.
Table 1- Please rearrange the table (CON and BAS should come as columns.) this will enhance the readability- better to use the format of Table 2
Response: We would like to acknowledge the comment of the reviewer, but in Table 1 we are reporting different experiments, which would result in a confusing and large table. We adopted this scheme as it has been approved in other review manuscript from our research group that was published in the same journal (Cappellozza et al., 2021). So, if this response is accepted, we would like to keep the format as it is.
Line 320- Please specify the animal here - Beef cattle?
Response: “beef steers” was included into the manuscript.
Line 322-324- not clear. Please rewrite elaborating the content.
Response: Rewritten as requested.
Table 2- Title should be revised- Please specify the beef animal - Beef cattle is better than beef animals
Response: Rewritten as requested and “newly-weaned beef steers” was included.
Table 3- rearrange the table as in Table 2 to enhance the readability. The title should be revised, please specify the animal as beef cattle. Define DMI and FE here.
Response: “Beef cattle” was included in the title of the Table 3.
Please check the comment above for Table 1, as the rationale used to create Table 3 was the same. DMI and FE were defined as requested.
Line 374- You should mention the species here - Beef cattle
Response: “Beef cattle” was included as requested.
Table 4- Please include the species in the title
Response: “Beef bulls” was included as requested.
Line 393- better to mention as beef cattle
Response: “Beef cattle” was included as requested.
Line 407- Define APR
Response: APR was already defined in line 107.
BAS vs Castration- too much information on the experimental methodology (unpublished data of your own research team). Better to include other references also here.
Response: The description of the experimental design and results have been adjusted. To the best of our knowledge, no other research work has been done with BAS in castrated animals.
Table 5- include the species in the title
Response: “crossbred beef steers” was included as requested.
Table 6- Include the species in the title
Response: “in finishing beef bulls” was included as requested.
In overall, too much of information in the introduction. Better to focus on your objectives and rearrange the introduction. Better to include more related literature to support your conclusion.
Better to organize the information in a more logical sequence to support your conclusion.
Response: This point has been discussed by the Editor and other reviewers and we highlighted that our research group has been leading the research efforts with BAS in ruminants. The other few work that has been done in beef and dairy cattle have also ben included into the manuscript.
We structured the review in a manner that would follow a logical rationale regarding stress, its physiology, and immunity to explain and set the ground on the subject and how pheromones could be important in reducing stress. Then, we finalize the review by providing the use of BAS in different production settings, supporting the idea that it can be used I different management strategies in beef and dairy cattle.
Round 2
Reviewer 1 Report
Dear authors, I appreciate you acknowledgement fo my concerns. As I said before, the article is interesting and well written. Especially with all the gerneral information on pheromons, it is more a book chapter than a scientific article. So I am sorry, to stick to my opion, but leave it to the editor to decide.
Author Response
Review article and a book chapter are essentially the same thing. However, a review article in a prestigious, high-profile journal will generally out-perform a book chapter in terms of citations and impact to the discipline.
Reviewer 2 Report
Dear Authors,
Thank you for taking into account my suggestions. The manuscript now reads smoothly, and the topic as said before is interesting. The “5.3. BAS vs. Castration” section has been improved too, although I still feel it should be addressed as original research to be published separately, and then cited in the present work. However, I leave it to the editor to decide on this aspect.
Author Response
Thanks! Data was previously published as abstract, and reference now included. That specific study did not contain enough data to warrant an independent journal publication.